# Evaluation of Glutaminase Expression in Prostate Adenocarcinoma and Correlation with Clinicopathologic Parameters

**DOI:** 10.3390/cancers13092157

**Published:** 2021-04-29

**Authors:** Zin W. Myint, Ramon C. Sun, Patrick J. Hensley, Andrew C. James, Peng Wang, Stephen E. Strup, Robert J. McDonald, Donglin Yan, William H. St. Clair, Derek B. Allison

**Affiliations:** 1Department of Internal Medicine, Division of Medical Oncology, University of Kentucky, Lexington, KY 40536, USA; p.wang@uky.edu; 2Markey Cancer Center, University of Kentucky, Lexington, KY 40536, USA; ramon.sun@uky.edu (R.C.S.); andrew.james@uky.edu (A.C.J.); Stephen.Strup@uky.edu (S.E.S.); donglin.yan@uky.edu (D.Y.); stclair@email.uky.edu (W.H.S.C.); Derek.Allison@uky.edu (D.B.A.); 3Department of Neuroscience, University of Kentucky College of Medicine, Lexington, KY 40536, USA; 4Department of Urology, University of Texas MD Anderson Cancer Center, Houston, TX 77030, USA; hpatrick1@mdanderson.org; 5Department of Urology, University of Kentucky, Lexington, KY 40536, USA; 6Department of Pathology and Laboratory Medicine, University of Kentucky, Lexington KY 40536, USA; rmcdonald@uky.edu; 7Department of Internal Medicine-Health Services Research, University of Kentucky, Lexington, KY 40536, USA; 8Department of Radiation Medicine, University of Kentucky, Lexington, KY 40536, USA

**Keywords:** glutaminase, immunohistochemistry, in situ methods, prostate, prognosis

## Abstract

**Simple Summary:**

High expression levels of glutaminase (GLS1) are reported for several cancers, and correlate with parameters of disease status. GLS1, the rate-limiting enzyme in the glutamine pathway, is involved in DNA/RNA and amino acid synthesis and contributes to other pathways (e.g., TCA cycle). Inhibition of GLS1 has shown anti-tumor activity in both solid tumors and hematological malignancies. The CB-839 agent, a novel GLS1 inhibitor, has been under investigation clinically. GLS1 expression by immunohistochemical (IHC) staining in prostate has not been definitively demonstrated. We present a retrospective study evaluating GLS1 expression utilizing The Cancer Genome Atlas (TCGA) RNA-Seq data and by IHC in formalin-fixed paraffin embedded radical prostatectomy samples. The study showed a significant difference in GLS1 levels between cancer and non-cancer, but fell short as a prognostic marker. As the study cohort was skewed to less aggressive localized prostate cancer, we support further studies that incorporate high-risk and very high-risk localized and metastatic prostate cancers.

**Abstract:**

High Glutaminase (GLS1) expression may have prognostic implications in colorectal and breast cancers; however, high quality data for expression in prostate cancer (PCa) are lacking. The purpose of this study is to investigate the status of GLS1 expression in PCa and correlated expression levels with clinicopathologic parameters. This study was conducted in two phases: an exploratory cohort analyzing RNA-Seq data for GLS1 from The Cancer Genome Atlas (TCGA) data portal (246 PCa samples) and a GLS1 immunohistochemical protein expression cohort utilizing a tissue microarray (TMA) (154 PCa samples; 41 benign samples) for correlation with clinicopathologic parameters. In the TCGA cohort, GLS1 mRNA expression did not show a statistically significant difference in disease-free survival (DFS) but did show a small significant difference in overall survival (OS). In the TMA cohort, there was no correlation between GLS1 expression and stage, Gleason score, DFS and OS. GLS1 expression did not significantly correlate with the clinical outcomes measured; however, GLS1 expression was higher in PCa cells compared to benign epithelium. Future studies are warranted to evaluate expression levels in greater numbers of high-grade and advanced PCa samples to investigate whether there is a rational basis for GLS1 targeted therapy in a subset of patients with prostate cancer.

## 1. Introduction

The essential role of dysregulated glucose metabolism, also known as “aerobic glycolysis”, in cancer was first discovered by Otto Warburg and colleagues in the 1920s [1,2,3]. Glucose and glutamine are essential nutrients for cell growth and survival [4]. Glutamine supplies nitrogen for purine and pyrimidine synthesis and nonessential amino acids for protein synthesis through glutaminolysis, which converts glutamine to glutamate by the rate-limiting enzyme, glutaminase (GLS). Subsequently, metabolic reactions produce alpha-ketoglutarate that contributes to the tricarboxylic acid (TCA) cycle and energy production in cancer cells through oxidative phosphorylation [5]. As such, glutaminolysis plays a significant role in the metabolic reprogramming of cancer cell growth and proliferation. It is now believed that glutaminolysis is associated with either activation of oncogenes (such as *MYC*) [6,7] and/or loss of tumor suppressor genes (such as *T**P53*) [8,9].

Cancer associated fibroblasts (CAFs) are associated with prostate cancer (PCa) growth, progression to metastasis, and the development of castration resistance [10,11,12,13]. Glutamine metabolism may play a role in CAFs by providing enhanced secretion of glutamine from the tumor stromal cells as demonstrated in an ovarian cancer animal model [14]. Several studies have shown that androgen receptor signaling increases the utilization of glycolysis and alters glutamine metabolism in prostate cancer (PCa) [15,16]. Thus, it was hypothesized that blocking glutamine metabolism would deprive cancer cells of needed nutrients and lead to cell death. Furthermore, a new first-in-class orally bioavailable glutaminase inhibitor (CB-839), which inhibits glutathione production and blocks tumor glutamine consumption, is in early phase clinical trials as monotherapy or in combination for select tumors; examples include colorectal cancer (ClinicalTrials.gov Identifier: NCT02861300), non-small cell lung cancer (ClinicalTrials.gov Identifier: NCT04250545), hematological malignancy (ClinicalTrials.gov Identifier: NCT03047993), brain cancer (ClinicalTrials.gov Identifier: NCT03528642), and so on. However, there are no current ongoing clinical trials for PCa at the time of this writing.

Several studies reported that increased GLS1 expression was associated with disease aggressiveness and could be used as a prognostic marker in select solid tumors [17,18,19,20]. However, there is a lack of high-quality studies analyzing GLS1 expression in PCa. As a result, in the present study, we report an exploratory cohort correlating mRNA GLS1 expression with disease-free and overall survival, followed by a tissue microarray (TMA) cohort comparing GLS1 immunohistochemical (IHC) protein expression in PCa tissue and benign prostatic tissue, as well as a correlation with a more in-depth set of clinicopathologic parameters.

## 2. Results

In the TCGA cohort, we analyzed mRNA expression levels for GLS1 from 246 prostate cancer specimens. GLS1 mRNA expression barely showed a statistically significant difference in overall survival (Figure 1A,B) and did not show a statistically significant difference in disease-free survival; however, there was a trend toward a worse disease-free survival with high GLS1 expression (Figure 1C,D).

To further analyze GLS1 expression and to validate these findings, we studied a cohort of 154 patients with localized prostate adenocarcinoma and performed GLS1 IHC, an in situ method, to specifically measure GLS1 protein expression in tumor cells and to correlate the expression with a larger number of clinicopathologic parameters. The clinicopathological features of prostate cancer cases in this cohort are presented in Table 1. Briefly, the ages ranged from 43 to 73 years with a mean range of 58 years. Most patients (78%) were Caucasian and 20% were Black. Half of the patients (52.9%) were stage T2; 43.1% were stage T3, and the majority (94.7%) were node negative. Regarding Gleason grade group, 42.2% had grade group 2, 18.8% grade group 3, 5.2% grade group 4, and 13.4% grade group 5. In this cohort, 53% were negative for GLS1 expression, 21.5% had low GLS1 expression, and 25.5% had high GLS1 expression. Ten patients had androgen deprivation therapy (ADT) exposure prior to definitive prostatectomy. Among them, 5 out of 10 were GLS1 0, 4 were GLS1 low, and 1 was GLS1 high. Essentially, 50% were GLS negative and 50% were GLS1 positive. There was no statistical difference in GLS1 expression in patients with prior ADT exposure prior to definitive prostatectomy.

To evaluate the clinical significance of GLS1 IHC protein expression in PCa, we compared the expression levels to benign glandular prostatic cells. A total of 41 cases of benign prostate cases were included; 17 out of 41 (41%) cases had low GLS1 protein expression and the remaining 24 cases (59%) had no observable GLS1 protein expression. IHC staining for GLS1 expression for both groups is shown in Figure 2. Additionally, we used the *t* test to compare the difference in GLS1 protein expression between malignant and benign prostate cases. We saw a statistically significant difference in GLS1 protein expression between PCa cells and benign glandular epithelium (*p* < 0.003) by the *t* test (Figure 3).

Furthermore, we compared the correlation between GLS1 expression and clinicopathological parameters as an indication of prognostic value in this cohort. There was no difference between GLS1 low vs. high protein expression for age, race, Gleason score, stage, node status, and smoking status by univariate analysis (Table 2).

In the TMA cohort, 55 out of 154 (36%) patients had biochemical progression, 65% underwent salvage radiation, and 35% had long-term hormonal therapy. Five patients (3%) had systemic chemotherapy in addition to hormonal therapy. The estimated 5-year biochemical recurrence-free survival rate for high, low, and no GLS1 expression was 67.2%, 64.0%, and 74.1%, respectively (*p* = 0.8). The median overall survival (OS) was 12 years (0–23 years). The median OS time for high, low, and no GLS1 expression were 10.8, 11.8, 14.5 years, respectively (*p* = 0.76). There was no biochemical PFS difference between no, low, and high GLS1 expression (*p* = 0.48) (Figure 4A). No statistically significant between GLS1 expression and OS (*p* = 0.76) was found (Figure 4B).

## 3. Discussion

High GLS1 expression has been reported as a potential prognostic marker for poor response in solid tumors such as colorectal and triple negative breast cancers [17,18]. In prostate cancer, previously published studies have reported that high GLS1 expression was associated with a higher Gleason score and higher tumor stage [21,22]. In order to explore whether GLS1 is a negative prognostic marker, we analyzed GLS1 RNA-Seq data from TCGA database. In this cohort, GLS1 mRNA expression did not show a statistically significant difference in disease-free survival but did show a small statistically significant difference in overall survival. We next sought to further validate these findings utilizing GLS1 IHC to restrict the analysis of GLS1 expression more specifically to tumor cells, and to study the relationship with a wider range of clinicopathologic parameters from our patient population. In summary, we found a lack of correlation between GLS1 protein expression by IHC and various clinicopathological parameters. However, we did find a statistically significant difference between GLS1 protein expression in PCa cells versus benign glands, with high GLS1 protein expression limited to prostate cancer. There are important differences between our study and previous investigations: (1) techniques/methods of measuring GLS1 expression; (2) tissue sample types (preclinical cell lines vs. patient radical prostatectomy samples); (3) patients’ baseline characteristics.

With respect to the first difference, Zhang et al. demonstrated that high expression of GLS mRNA was significantly associated with high Gleason score (≥8) and higher tumor stage (≥T3) [22]. They utilized qRT-PCR and Western blot analyses to measure GLS mRNA while we analyzed RNA-Seq data. Furthermore, no in situ methods were performed, such as IHC, which is important for prostate cancer considering tumors are often rich in stroma and intimately associated with benign glands. In contrast, Dorai et al. studied GLS IHC on FFPE samples of PCa [23]. However, a careful examination of their GLS IHC figures shows that staining was primarily localized to the stroma and not the actual tumor cells, which questions the reliability of the data [23]. Prior to the present study, a high-quality in situ investigation that localizes GLS1 expression to tumor cells, the biologically relevant compartment, has been lacking. In our study, we used a combination of RNA-Seq data, as well as a high quality in situ IHC protein method, to study GLS1 expression (Figure 3).

With respect to the second difference, the Zhang et al. study also reported that the expression of GLS1 mRNA levels by qRT-PCR and Western blot methods were higher in different prostate cancer cell lines (DU145, PC-3 and LNCaP) compared with a normal prostate epithelial cell line (RWPE-1) [22]. This finding is consistent with our IHC results, which showed a statistically significant difference in GLS1 protein expression in prostate cancer cells versus benign glandular epithelium.

Finally, with respect to the third difference, many of our samples in the TMA cohort were from patients with low/intermediate risk, localized prostate cancer: T2 (52.9%), Gleason grade group 2 (42.2%), Gleason grade group 5 (13.4%) and N0 (94.7%). Furthermore, the TCGA cohort is similarly enriched for localized, low-to-intermediate grade PCa. In contrast to our study, the Zhang et al. study included more high-risk patients; Gleason score ≥8 (63%), ≥T3 (56%) and PSA ≥10 ng/mL (59%) [22]. Given the smaller sample size for aggressive disease, our cohorts may not be sufficiently powered to detect a difference in this population. Furthermore, the cell lines DU145, PC-3, and LNCap are all derived from Stage IV metastatic prostate cancer. As a result, it is not surprising that GLS1 expression may be different in these aggressive tumors than in our cohorts. To this point, Zacharias et al. evaluated the metabolic differences between the aggressive prostate cancer cell line (PC3) and the even more aggressive metastatic cell line (P3M) by using hyperpolarized in vivo pyruvate studies, nuclear magnetic resonance spectroscopy, and carbo-13 feeding studies, showing that the P3M cell line utilized a higher amount of glutamine than PC3 [24]. These findings are interesting, and more studies are warranted in high-risk and aggressive PCa.

In addition, GLS exists in two major isoforms, kidney-type glutaminase (KGA) and glutaminase C (GAC) [25], which are collectively referred to as GLS1; the antibody used for these analyses detects both isoforms [26]. Since GAC may be more active than KGA, further studies that utilize in situ methods of detection with these two splice variants may be warranted. However, interestingly, Zacharias et al. performed Western blot analyses of KGA and GAC and showed similar expression levels in all prostate cancer cell lysates for both the PC3 and PC3M cell lines [24]. As a result, since our antibody detects both KGA and GAC, it is unclear that additional IHC would provide any further information. To this point, GLS1 expression (detecting both KGA and GAC) was found to be undetectable in 53% of our prostate cancer samples. This finding suggests that KGA specific and GAC specific IHC would be negative in the majority of our samples. Despite lack of any additional IHC markers, our data suggest that the glutaminolysis pathway may not play a major role in many clinically encountered, localized prostate cancers. However, additional studies are warranted, particularly in more high-risk and advanced prostate cancer cohorts.

Another limitation in our TMA cohort is related to the age of the FFPE samples, which ranged from 4–18 years old. Antigenicity (antibody to antigen specificity) can change over time; however, it is crucial to include samples from patients for whom long-term follow-up is available. This factor may result in less intense staining; however, in comparison to the published literature, our rate of GLS1 protein expression in prostate cancer is not dramatically dissimilar from mRNA data. One study reported that GLS1 expression was found in 64% (68 out of 107 patients) in the malignant prostate specimens by mRNA analysis [21], while in our study, GLS1 IHC protein expression was found in 47% (70 out of 154 patients) malignant prostate specimens.

Despite the lack of support for GLS1 IHC protein expression as a prognostic marker in early stage prostate cancer in our study, the findings from preclinical models and the trend in our TCGA analysis warrant further investigation into whether GLS1 expression could be a predictive marker for response to the CB-839, a small molecule allosteric inhibitor of GLS. Furthermore, the fact that high protein expression by IHC was restricted to cancer specimens may indicate that this pathway could be a potential target for a subset of patients. Importantly, pre-clinical studies have shown anti-tumor activity in PCa cell lines treated with CB-839 as monotherapy [24], and it showed synergistic activity in combination with the PARP inhibitors in multiple solid tumors including PCa cells [27], in combination with metformin in osteosarcoma [28] in vitro, in combination with cabozantinib in renal cell carcinoma [29], in combination of CDK4/6 inhibitor in colorectal, and breast cancers preclinical models [27] as well as CB-839 could be utilized as a radiosensitizer in lung cancer shown in vivo and vitro [30].These encouraging in vitro results support the confirmation of CB-839 anti-tumor activity in xenograft models, but these findings need to be further translated. Essentially, additional studies enriched for high-risk and aggressive PCa are warranted.

## 4. Materials and Methods

This study was conducted in two phases: an exploratory cohort analyzing RNA-Seq data for GLS1 from The Cancer Genome Atlas (TCGA) data portal and a GLS1 immunohistochemical (IHC) protein expression cohort utilizing a TMA for correlation with additional clinicopathologic parameters.

### 4.1. TCGA Cohort

246 PCa samples with RNA-Seq data available for GLS1 were identified from TCGA (https://cancergenome.nih.gov/, accessed on 4 March 2021) and included in an exploratory phase of this study.

### 4.2. TMA Cohort

A TMA was constructed from formalin-fixed paraffin-embedded tissue from 154 patients who underwent radical prostatectomy for localized prostate adenocarcinoma between 2002 and 2016 at University of Kentucky Markey Cancer Center. Survival intervals were calculated from the time of prostate surgery until death or last clinic follow-up. Clinicopathological information including age, race, tumor stage, histology, grade, lymph node status, Gleason score, smoking status, subsequent treatment history (radiation therapy and or hormonal therapy), and survival data were collected using the Kentucky Cancer Registry. The study was approved by the local Institutional Review Boards (IRBs) at University of Kentucky (53854).

The TMAs were used to evaluate GLS1 protein expression by IHC. Slides were sectioned at 4 microns and baked at 58 °C for a minimum of one hour. Staining was conducted on the Ventana Discovery Ultra using Standard CC2 (Roche, Tucson, AZ, USA) antigen retrieval, followed by incubation with an anti-GLS1 antibody (ab156876, abcam, Cambridge, MA, USA) at 1:400 for 1 h at 37 °C and then incubation with OmniMap anti-Rabbit-HRP (Roche, Tucson, AZ, USA) and visualization with DAB (Roche), according to the manufacturer’s recommendations. Slides were lightly counterstained in Mayer’s hematoxylin before permanent mounting. Two board-certified anatomic pathologists participated in this study (Derek B. Allison and Robert J. McDonald) and worked together to optimize the IHC staining conditions and agree upon staining thresholds. Cases where there was a question about scoring were reviewed by both pathologists. The expression scores were calculated as intensity score (0 = no staining; 1 = weak granular cytoplasmic staining; 2 = moderate granular cytoplasmic staining; 3 = strong granular cytoplasmic staining) X proportion score (0 = no positive cells; 1= <10% positive cells; 2= 10–40% positive cells; 3 = 40–70% positive cells; 4 = >70% positive cells, as modified from Xiang et al [31]. For the PCa samples, interpretation was restricted to the PCa tumor cells while interpretation in the benign tissue samples was restricted to the benign glandular epithelium. Based on the distribution of the scores, the cases were then classified into one of three tiers: no expression, low expression, or high expression. Forty-one benign prostate samples were included for comparison and scored using the same method. Associations between GLS1 levels and clinicopathological parameters and survival were analyzed by Pearson’s chi-squared and Log-rank tests.

### 4.3. Statistical Analysis

#### 4.3.1. TCGA Cohort

Kaplan–Meier estimate or survival analysis was performed using the publicly available software GEPIA (http://gepia.cancer-pku.cn, accessed on 4 March 2021) that utilizes RNA sequencing expression data from tumor and normal samples from the TCGA database using a standard processing pipeline. For both disease-free survival and overall survival, upper and lower quartile were used for high and low expression cut-offs and the nonparametric Mantel-Cox test was used for the calculation of Log-rank score.

#### 4.3.2. TMA Cohort

A total of 154 patients were included in the TMA cohort IHC analysis. Basic characteristics, including age, race, smoking history, pathological tumor stage, pathological node status, Gleason grade, and GLS expression were summarized by descriptive statistics. Univariate analysis was conducted to examine differences in age, race, Gleason score, stage, node status, and smoking status by GLS expression (no expression, low expression, or high expression). Associations between categorical endpoints were examined by Chi-square test, and continuous endpoints were compared by Student’s *t*-test. Disease-free (biochemical recurrence-free) survival and overall survival were estimated and plotted using the Kaplan–Meier method. Log-rank tests were performed to detect survival differences by GLS expression. Analyses were conducted using SAS 9.4 (SAS Institute Inc., Cary, NC, USA) and R Studio 1.4 (RStudio, PBC, Boston, MA, USA). All hypotheses testing and confidence intervals were conducted at 95% significance level.

## 5. Conclusions

In our study, PCa cells were more likely to have increased GLS1 protein expression compared to benign glandular epithelium. Although GLS1 protein expression did not appear to be a statistically significant prognostic marker, high GLS1 mRNA expression showed a trend toward worse disease-free and overall survival. Our cohorts, however, were enriched for cases with localized disease and low-to-intermediate grade PCa. As a result, future studies are warranted to evaluate GLS1 expression in high grade and advanced PCa cases to determine whether there is a rational basis for GLS1 targeted therapy with CB-839 in a subset of patients with PCa.

## Figures and Tables

**Figure 1 cancers-13-02157-f001:**
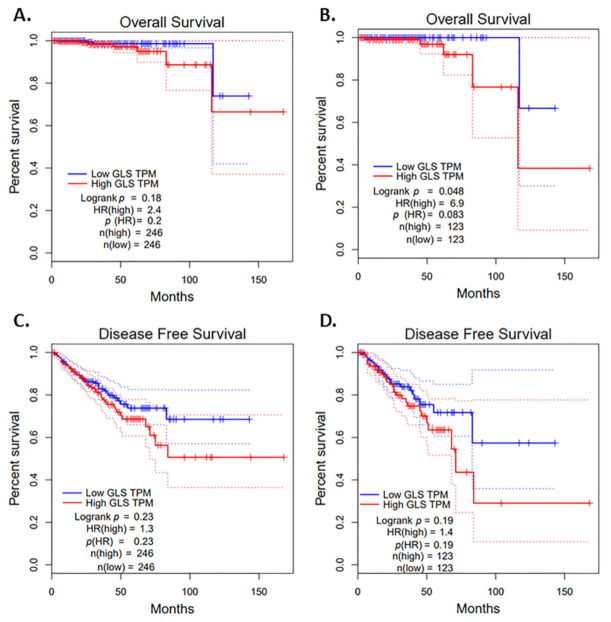
Impact of glutaminase (GLS1) mRNA on survivals using The Cancer Genome Atlas (TCGA) data portal cohort. (**A**) GLS1 mRNA expression and overall survival with a medium cutoff, (**B**) GLS1 mRNA expression and overall survival with a quartile cutoff, (**C**) mRNA expression and disease-free survival with a medium cutoff and (**D**) GLS1 mRNA expression and disease-free survival with a quartile cutoff. Dotted lines represent 95% confidence interval (CI).

**Figure 2 cancers-13-02157-f002:**
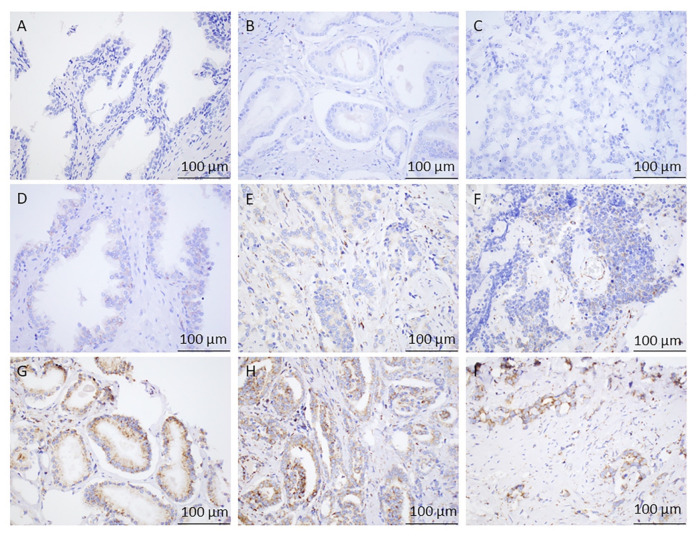
Immunohistochemical Analysis of GLS1 in Prostate Tissues. (**A**) Benign central zone of the prostate with no glutaminase (GLS1) staining, 40× magnification. (**B**) Grade Group 1 Prostate Adenocarcinoma showing no staining with GLS1, 40× magnification. (**C**) Grade Group 3 Prostate Adenocarcinoma showing no GLS1 expression, 40× magnification. (**D**) Benign Central zone of the prostate with faint but diffuse granular cytoplasmic staining (low expression) with GLS1, 40× magnification. (**E**) Grade Group 2 Prostate Adenocarcinoma showing faint but diffuse granular cytoplasmic staining (low expression) with GLS1, 40× magnification. (**F**) Grade Group 5 prostate adenocarcinoma showing moderate but focal granular cytoplasmic staining (low expression) with GLS1, 40× magnification. (**G**) Grade Group 1 Prostate Adenocarcinoma showing strong and diffuse granular cytoplasmic staining (high expression) with GLS1, 40×. (**H**) Grade Group 3 Prostate Adenocarcinoma showing variably strong but diffuse cytoplasmic granular staining (high expression) with GLS1, 40×. (**I**) Grade Group 5 Prostate Adenocarcinoma showing single infiltrating tumor cells and poorly formed glands with strong, diffuse granular cytoplasmic staining (high expression) with GLS1, 40×.

**Figure 3 cancers-13-02157-f003:**
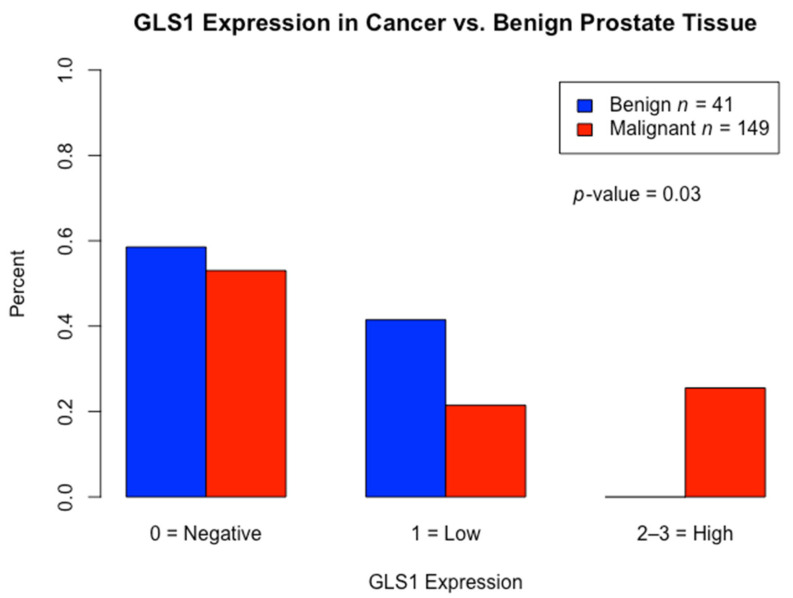
Distribution of glutaminase (GLS1) expression: Comparison of GLS1 protein expression between malignant prostate cancer vs. benign prostate tissue (control) by *t*-test.

**Figure 4 cancers-13-02157-f004:**
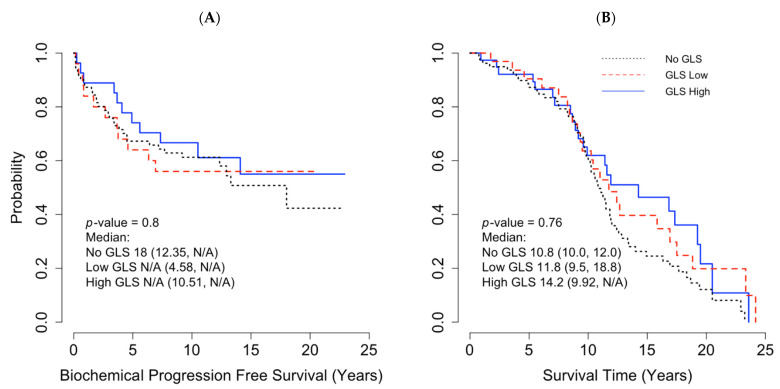
Impact of glutaminase (GLS1) mRNA on survivals with using a tissue microarray (TMA) Cohort. (**A**) Pearson correlation between GLS1 protein expression and biochemical progression-free survival and (**B**) Pearson Correlation between GLS1 protein expression and overall survival. N/A = estimates not available due to small sample size/censoring.

**Table 1 cancers-13-02157-t001:** Clinicopathological characteristics of prostrate adenocarcinoma cases in tissue microarray (TMA) cohort.

Clinicopathological Characteristics	Number of Prostate Cancer Cases, *n* (%)
	Total *n* = 154
Age	
<60	84 (55%)
≥60	65 (45%)
Race	
White	121 (78.6%)
Black	31 (20.1%)
Other	2 (1.3%)
Smoking	
Yes	56 (36.4%)
No	50 (32.5%)
Unknown	48 (31.2%)
Pathological Tumor (pT) stage	
2	81 (52.9%)
3	66 (43.1%)
4	6 (3.9%)
Missing	1
Pathological Node (pN) status	
Yes	7 (5.3%)
No	124 (94.7%)
Missing	23
Gleason Grade Group	
Grade Group 1	21(13.6%)
Grade Group 2	65(42.2%)
Grade Group 3	29(18.8%)
Grade Group 4	8(5.2%)
Grade Group 5	21(13.6%)
Missing	10 (6.4%)
GLS1 Score	
High expression	38 (25.5%)
Low expression	32 (21.5%)
Negative	79 (53%)
Missing *	5

* Missing means there was no remaining tumor cells available for scoring in the prostate cancer (PCa) sample.

**Table 2 cancers-13-02157-t002:** Glutaminase (GLS1) immunohistochemistry protein expression and clinicopathological parameters of prostate adenocarcinoma cases in TMA cohort.

Variables	GLS1 Score				*p*-Value
	High Expression (*n* = 38)	Low Expression (*n* = 32)	No (*n* = 79)	Missing (*n* = 5)	
Age					0.29
<60	20 (24%)	17 (20%)	47 (56%)		
≥60	18 (28%)	15 (23%)	32 (49%)		
Race					0.3
White	33 (86.8%)	27 (84.4%)	57 (72.2%)	4	
Black	5 (13.2%)	5 (15.6%)	20 (25.3%)	1	
Other	0	0	2 (2.5%)		
pT stage					0.6
pT2	19 (50%)	18 (56.3%)	40 (51.3%)	4	
pT3	19 (50%)	12 (37.5%)	34 (43.6%)	1	
pT4	0	2 (6.3%)	4 (5.1%)	0	
Missing	0	0	1		
pN					0.2
Yes	1 (2.9%)	0	5 (7.9%)	1	
No	34 (97.1%)	29 (100%)	58 (92.1%)	3	
Gleason Grade Group					0.5
1	9 (23.6%)	6 (18%)	5 (6.3%)	1	
2	16 (42%)	11 (34%)	37 (46.8%)	1	
3	4 (10.5%)	6 (18.7%)	18 (22.8)	1	
4	3 (7.8%)	1 (3%)	4 (5%)		
5	5 (13%)	5 (15.6%)	11(13.9%)		
Unknown	1(2.6%)	3 (9.3%)	4 (5%)	2	
Smoking					0.9
Yes	12 (31.6%)	13 (40.6%)	28 (35.4%)	3	
No	13 (34.2%)	10 (31.3%)	27 (34.2%)	0	
Unknown	13 (34.2%)	9 (28.1%)	24 (30.4%)	2	

pT = pathological stage, pN = pathological node.

## Data Availability

The data presented in this study are available on request from the corresponding author.

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
