# Peer review of "Evaluation of Glutaminase Expression in Prostate Adenocarcinoma and Correlation with Clinicopathologic Parameters"

_cancers, 2021, doi:10.3390/cancers13092157_

Round 1

Reviewer 1 Report

Overall, the work is sound and provides worthwhile additional knowledge to the biomedical community, even if the prognostic findings of GLS1 expression in PCa are not strong. The following suggestions will improve the article.

Figures 1 and 2 could be combined (Fig 1a, b etc). Indeed, the tool used to generate K-M curves (GEPIA) could be utilsed further to generate K-M curves with a median cutoff (or even custom cutoffs), and this additional analysis would provide more insight into the effects of GLS1 expression on survival.

Likewise, Figures 5 and 6 could be combined. They don't need to be standalone figures.

Figure 3: is lacking scale bars on the images. These should be added.

Figure 4: it would be more useful if the data points are distributed so the reader can see how many samples are at each expression score.

In line 209 the authors “anti-GLS1 monoclonal antibody (CB-839)”. CB-839 is not a monoclonal antibody, it is an small molecule allosteric inhibitor of GLS, as the authors state earlier in the article.

Methods on line 225 and following paragraph (titled TCGA Cohort) detail the same information as in the paragraph starting on line 262. This duplication makes it more difficult to ascertain what the authors actually did. I suggest deleting the first section.

The grammar used is mostly OK but some minor errors are present and the manuscript would benefit from proofreading. In one part of the article present rather than past tense is used.

Line 21: “hematology malignancies” should be “hematological malignancies”

Line 22: “GLS1 inhibitor, a novel CB-839 agent,” should be “CB-839, a novel GLS1 inhibitor,”

Line 58: P53 should be TP53

Author Response

Thank you very much for your review. Below are our responses in bold.  

Figures 1 and 2 could be combined (Fig 1a, b etc). Indeed, the tool used to generate K-M curves (GEPIA) could be utilsed further to generate K-M curves with a median cutoff (or even custom cutoffs), and this additional analysis would provide more insight into the effects of GLS1 expression on survival.

We added KM curves with a median cutoff in addition to quartile cutoff. We combined all figs into one combined.

Likewise, Figures 5 and 6 could be combined. They don't need to be standalone figures.

We combined Figs 5 and 6 together.

Figure 3: is lacking scale bars on the images. These should be added.

We added scale bars on Fig 3.

Figure 4: it would be more useful if the data points are distributed so the reader can see how many samples are at each expression score.

We have updated the Fig as suggested.

In line 209 the authors “anti-GLS1 monoclonal antibody (CB-839)”. CB-839 is not a monoclonal antibody, it is an small molecule allosteric inhibitor of GLS, as the authors state earlier in the article.

We have edited as suggested.

Methods on line 225 and following paragraph (titled TCGA Cohort) detail the same information as in the paragraph starting on line 262. This duplication makes it more difficult to ascertain what the authors actually did. I suggest deleting the first section.

We deleted the duplicated information in the TCGA Cohort under Methods section.

The grammar used is mostly OK but some minor errors are present and the manuscript would benefit from proofreading. In one part of the article present rather than past tense is used.

The manuscript has been proof-read.

Line 21: “hematology malignancies” should be “hematological malignancies”

We have edited as suggested

Line 22: “GLS1 inhibitor, a novel CB-839 agent,” should be “CB-839, a novel GLS1 inhibitor,”

We have edited as suggested.

Line 58: P53 should be TP53

We have edited as suggested.

Reviewer 2 Report

This manuscript is very well written. The authors present a sound hypothesis and the design of the study is appropriate to test that hypothesis. The conclusions are fully supported by the results. The only suggestion I have is if the authors can include the hazard ratio and 95% confidence intervals in the figures in addition to the provided logrank p-value.

Author Response

Thank you for your review. Below is our response:

We have added hazard ratio and 95% CI on Fig 1. We added 95% CIs for the median survival time in both survival figures in addition to the Logrank -p value. We didn’t include the pair-wise HRs because the overall p-value is not significant, so we think the observed median survival time and corresponding CIs might be more informative for the readers and future studies. 

Reviewer 3 Report

Based on metabolic dis-regulations in cancer progression, the presented study could be really interesting. The main topic of the manuscript is to study glutaminase-1 expression in prostate cancer tissues in situ and evaluate its clinic-pathologic importance. The problem is that the high quality data related to the expression of GLS1 is still missing after your work.

After evaluating RNA expression data sets where GLS1 mRNA expression correlated to OS and DFS in prostate cancer cases, the authors have analysed 154 PCa specimens using TMA and immunohistochemistry. This sample number allows an impressive study using clinical data with long follow up (more than 12 years). Nevertheless, the antibody which was selected in their work could not distinguish the different isoforms of GLS1 and the authors forgot to mention in their introduction that these isoforms can have special, potentially different functions both in tumour progression and glutaminase inhibitor sensitivity. Based on these, it is not surprising that the authors could not find any correlations in their cohorts between GLS1 expression, patients survival and other data. It could be that it is true, that there are no any correlations, but such a negative statement and its high quality value need rather precise analysis and comprehensive study.

Major suggestion:

  • It is necessary to perform more IHC studies with other specific antibodies which distinguish KGA and GAC GLS1 isoforms in situ and additionally it would be advisable analysing the tissue heterogeneity, as well. To study the different isoforms in situ has special importance because it is well-known that prostate cancer cell lines express both isoforms (e.g. it was presented in Zacharias study in Western blot or in many other works).

Other comments

  • the evaluation of the low and high scores must be precisely given, especially because the presented low and high scored IHC photo documentations have different magnifications and intensive positive staining except for negative case.
  • Was the tissue heterogeneity or staining heterogeneity considered in their evaluation?
  • Compering with Fig3, the results in Fig4 cannot be interpreted. According to the current presentation, it seems that almost all cases were between 0-1 score. Moreover, Fig4 needs a more detailed legend/description.

Based on these, I suggest reviewing more carefully the previously published studies (there are many experimental and pathology studies) and revising your work, performing more IHC stainings and involving more pathologists in staining evaluations. To accept this manuscript, it needs additional work and major/substantial revision.

Author Response

Major suggestion:

  1. It is necessary to perform more IHC studies with other specific antibodies which distinguish KGA and GAC GLS1 isoforms in situ and additionally it would be advisable analysing the tissue heterogeneity, as well. To study the different isoforms in situ has special importance because it is well-known that prostate cancer cell lines express both isoforms (e.g. it was presented in Zacharias study in Western blot or in many other works).

Thank you for your review. We agree completely that there needs to be additional studies performed in order to fully understand and grasp the complexity of glutamine metabolism in prostate cancer. These future studies, in combination with the present one, will hopefully uncover a role for CB839 for a subset of patients with prostate cancer. Your point regarding the GLS1 antibody utilized for IHC is valid and perhaps is a good future direction for additional studies in high risk, aggressive human prostate cancer samples, as is its relation to androgen receptor signaling. We would like to point out, however, that the western blots in the referenced Zacharias study showed that the levels of GAC and KGA GLS1 were similar in all prostate cancer cell lysates in both the PC3 and PC3M cell lines. As a result, since our antibody detects both GAC and KGA together, it is unclear what significance this additional IHC would add to the current study. Furthermore, GLS1 expression (detecting both GAC and KGA) was found to be undetectable in 53% of our prostate cancer samples, which is data worth reporting. This finding means in combination, GAC and KGA expression is below the limit of detection by IHC in 53% of routinely encountered prostate cancer. As a result, we already know that separate GAC- and KGA-specific IHC would be negative in this entire group. In summary, we feel that our current results are significant enough to be published with the understanding that additional studies are always warranted.

We added the paragraph below in the discussion section from line 223 – line 238.

“In addition, GLS exists in two major isoforms, kidney-type glutaminase (KGA) and glutaminase C (GAC), which are collectively referred as GLS1; the antibody used for these analyses detects both isoforms. Since GAC may be more active than KGA, further studies that utilizing in-situ methods of detection with these two splice variants may be warranted. However, Zacharias et al. performed western blot analyses of KGA and GAC and showed similar expression levels in all prostate cancer cell lysates for both the PC3 and PC3M cell lines. As a result, since our antibody detects both KGA and GAC, it is unclear that additional IHC would provide any further information. To this point, GLS1 expression (detecting both KGA and GAC) was found to be undetectable in 53% of our prostate cancer samples. This finding suggests that KGA specific and GAC specific IHC would be negative in the majority of our samples. Despite lack of any additional IHC markers, our data suggest that the glutaminolysis pathway may not play a major role in many clinically-encountered, localized prostate cancers. However, additional studies are warranted particularly in more high-risk and advanced prostate cancer cohorts.”

  1. The evaluation of the low and high scores must be precisely given, especially because the presented low and high scored IHC photo documentations have different magnifications and intensive positive staining except for negative case. Was the tissue heterogeneity or staining heterogeneity considered in their evaluation?

Variable expression and heterogeneity are always important to take into account when evaluating protein expression. As a result, it is common convention to utilize a scoring system that takes into account the intensity of staining and the percentage of cells that stain with that particular intensity. These are then added together to get a combined score that holistically accounts for variations in expression. This method is well-accepted for the purposes of dealing with staining heterogeneity and is what has also been suggested for GLS1 staining in other organ systems, as we reference in our methods. Please note this text in our methods section: “…calculated as intensity score (0= no staining; 1= weak granular cytoplasmic staining; 2= moderate granular cytoplasmic staining; 3= strong granular cytoplasmic staining) X proportion score (0=no positive cells; 1= <10% positive cells; 2= 10–40% positive cells; 3=40–70% positive cells; 4= >70% positive cells, as modified from Xiang et al.13” In order to deal with the complexity of these data, it is common convention to then segment expression based on the calculated scores. Please see the following text in the methods section: “Based on the distribution of the scores, the cases were then classified into one of three tiers: no expression, low expression, or high expression.” Again, this is common convention for IHC interpretation for research and for many clinical purposes. In addition, two board-certified anatomic pathologists participated in this study (DBA & RJM) and worked together to optimize the IHC staining conditions and agree upon staining thresholds. Cases where there was a question about scoring were reviewed by both pathologists. 

We added line 295 to line 298 in methods section as below:

“Two board-certified anatomic pathologists participated in this study (DBA & RJM) and worked together to optimize the IHC staining conditions and agree upon staining thresholds. Cases where there was a question about scoring were reviewed by both pathologists.

  1. Compering with Fig3, the results in Fig 4 cannot be interpreted. According to the current presentation, it seems that almost all cases were between 01- score. Moreover, Fig 4 needs a more detailed legend/description.

We agree with your critique of the images selected for Figure 3 (now Fig 2). As a result, we have created a new Figure with additional images and a more detailed caption. We have updated Fig 4 (now Fig 3).